



# Dynamic Response of Offshore Wind Turbine Structure under Multi-load Coupling Based on DEM and FEM Joint Analysis

Xin GUAN[1*], Haoran XU[1], Ying YUAN[2], Shuaijie WANG[1], Chenhao ZHAO[1], Hua YU[1]

*1 School of New Energy, Shenyang Institute of Technology, Shenyang 110136, China;*

*2 China Quality Certification Center, Beijing FengTai 100000*

*Corresponding author Email: xin_guan@sina.com*

**Abstract**:The structural dynamic characteristics of offshore wind turbines are directly related to the operational safety and equipment reliability of these turbines in service. However, due to the complex working conditions, a single load analysis fails to accurately reflect the structural dynamic characteristics during actual operation. In this study, we focus on the 5MW offshore wind turbines and establish a three-dimensional turbulent flow field model at sea using the Kaimal wind speed spectrum. Additionally, we incorporate the Kärnä ice force spectrum to develop a mathematical model for floating ice. By combining multiple working conditions through permutation and combination techniques, we replicate the actual operating environment of offshore wind turbines. Leveraging OpenFAST's open computing capabilities and EDEM's discrete element analysis method, we investigate the dynamic response characteristics of wind turbines under separate and coupled effects of wind load, wave load, and ice load across different offshore working conditions. Our findings indicate that under coupling effects from wind-wave-ice loads, lateral and longitudinal displacement at the tower top as well as lateral and longitudinal bending moment at the tower foundation are greater compared to individual loads; however, cumulative fatigue damage caused by coupling loads on wind turbines is less than that resulting from individual loads.

**Keywords**:Offshore wind turbine, wind-wave-ice load, coupling, dynamic response

## 0 Introduction

In recent years, China has witnessed the extensive construction of offshore wind farms. The safe running of offshore wind turbines in the northern sea is affected by intermittent floating ice and turbulent winds, which are unique working conditions they must confront. The intermittent impact of floating ice presents a distinctive marine environment challenge for offshore wind power development, as it induces long-lasting and relatively stable vibrations on the overall structure due to ice load coupled with ocean current movement (LIU Weimin et al., 2019 and ZHANG Dayong et al., 2018). Such vibrations can lead to fatigue damage and continuous structural strain accumulation under sustained exposure to ice load. Additionally, the wind load exerts significant vibrational effects on both the tower foundation and mud surface line position of the unit's platform structure. These loads and their coupling effects alter aeroelasticity and structural dynamic response of wind turbines, making it crucial to investigate their influence on operational status under different marine environmental conditions. Considering various installed capacities and structural types of wind turbines nowadays, it becomes evident that annual ice formation characteristics in the northern Gulf of China pose a substantial threat to operational safety in this region (HUANG Yan et al., 2016). Therefore, studying wind load, ice load, wave load along with their interactions holds great engineering significance for advancing offshore wind power development in China.

Shi Wei et al. (2021) employed ANSYS/AQWA to establish a hydrodynamic numerical model for a semi-submersible floating foundation and conducted dynamic response analysis of a 10 MW offshore floating wind





turbine under the influence of wind and waves. Hu Xuan et al. (2019) developed the Kane multi-body dynamic
model for wind turbines, utilized the asynchronous Matlock model to simulate ice-induced vibration, and
investigated the tower's dynamic response to turbulent wind loads and asynchronous ice-induced vibrations.
Yang Dongbao et al. (2021) employed a discrete element model with bond-crushing capability to characterize
the damage and failure behavior of level sea ice, while utilizing the DEM-FEM coupling approach to simulate the
interaction process between single-pile wind turbines and level ice under varying ice velocities and thicknesses.
Wang Guojun et al. (2022) conducted a statistical analysis on the relationship between sea ice fracture length and
sea ice thickness, and utilized a modified deterministic ice force function to calculate the structural response of the
ice under measured conditions. Huang Yan et al. (2016) simulated the dynamic characteristics of wind power
structures with significant differences in their master-slave structure, and performed transient dynamic analysis
throughout the entire time domain. Zhang Lixian et al. (2019) carried out dynamic response analysis on single-pile
offshore wind turbines subjected to floating ice using FAST coupling numerical analysis. Ba Yueqiao (2019)
conducted fatigue analysis on offshore wind turbines in icy areas, and proposed a discrete element analysis
method for determining ice loads on offshore wind turbines.
Based on ABAQUS finite element numerical analysis software and FAST numerical analysis software,
Heinonen et al. (2008) conducted a dynamic characteristic analysis of single-pile wind turbines under the action of
floating ice. The results showed that joint mathematical model analysis could improve the reliability of offshore
wind turbine structural design. Wang et al. (2015) analyzed the fatigue characteristics influencing factors (ice load
and wind load) of offshore wind turbines using Kärnä ice force spectrum model; Wei Shi et al. (2016) combined
with semi-empirical structural action model of sea ice to analyze the dynamic response of single-pile wind turbine
structures under different ice velocities and thicknesses' actions by ice cones. Matlock et al. (1971) added spring
damping during loading process modeling analysis for floating ice, while Kärnä et al. (1999) proposed self-excited
vibration response calculation method for marine structures based on actual measurement data from sea ice, which
was calculated with sawtooth-shaped ice force function.
Currently, there is a limited number of studies investigating the impact of multi-load and ice load coupling on
the fatigue life and safe running of offshore wind turbines. Considering the presence of floating ice in the northern
seas of China and the national economic development's demand for offshore wind power, it is imperative to
examine the dynamic response of offshore wind turbines under the combined influence of multi-load and ice load.
In this study, numerical analysis using discrete element method and turbulent spatial coherence model was
conducted to investigate the dynamic response of NREL 5 MW offshore wind turbines subjected to wind-wave-ice
load coupling. The effects of different durations, ice excitation, and coupled load excitation on ultimate dynamic
response and fatigue life were explored. Additionally, frequency domain response analysis was performed for
large-scale offshore wind turbines under single action from ice load as well as coupled excitation from multiple
loads. Cumulative effect on structural fatigue response due to various combinations of fatigue damage was





considered, while relative errors in fatigue loading caused by individual actions (wind-wave-ice) versus coupling
effects were calculated. These findings aim to provide valuable insights for future construction projects involving
offshore wind power in cold regions within China.
**1 Load Calculation**
**1.1 Calculation Theory of OpenFAST Wind Load**
The aerodynamic load calculation of wind turbines primarily relies on the blade element momentum theory
and the generalized dynamic inflow theory to accurately describe wind loads. In this study, we employ the blade
element momentum theory to discretize wind turbine blades during load calculations. By iteratively calculating
stress and moment acting on each blade element along the span, a closed-loop solution is established between lift
and resistance coefficients and the induction factor. This approach ensures both engineering applicability and
calculation accuracy.
(1) Kaimal turbulent wind spectrum model
The turbulent wind spectrum mathematical model is established using the Sandia method in TurbSim
(VERITAS D N, 2010), and is generated based on the time series of Kaimal wind spectrum (IEC 61400-1, 2005) . The
longitudinal ($u$), lateral ($v$), and vertical ($w$) directions of the fluid domain are defined with respect to the tower
top coordinate system of the wind turbine, as illustrated in Figure 1

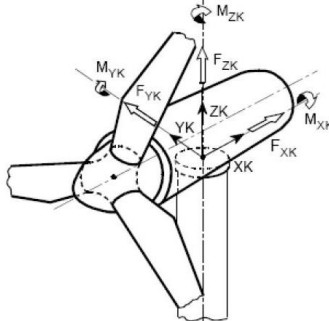


Fig. 1 Coordinate system at the top of the wind turbine tower
The mathematical formulation of the fluctuating wind velocity spectrum is presented.

$$S_K(f) = \frac{4\sigma_K^2 L_K / \overline{u}_{hub}}{\left(1 + 6 f L_K / \overline{u}_{hub}\right)^{\frac{5}{3}}} \tag{1}$$

Where, K represents the directions of the fluid domain, namely $u$, $v$, and $w$; $f$ denotes the circulation frequency; $L_K$
is defined as the integral scale parameter.



$L_K = \begin{cases} 8.10\Lambda_U, & K = u \\ 2.70\Lambda_U, & K = v \\ 0.66\Lambda_U, & K = w \end{cases}$ , $\Lambda_U$ is turbulence scale parameter, $\Lambda_U = 0.7\min(60m, HubHt)$, HubHt is the hub
height. The standard deviation of different directions is defined as: $\sigma_v = 0.8\sigma_u$, $\sigma_w = 10.5\sigma_u$, and the turbulent
wind load spectrum at the wheel hub of the wind turbine is formed, as shown in Figure 2.

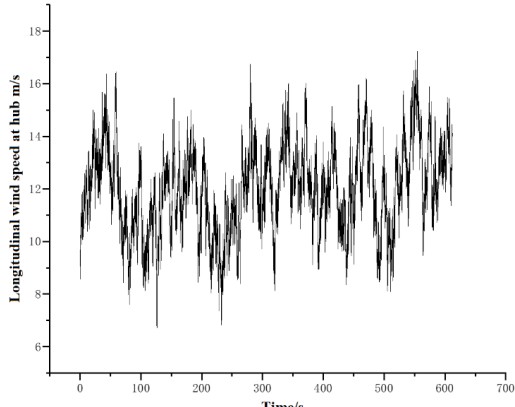


Fig. 2 Turbulent wind load spectrum at t hub (Kaimal wind spectrum)
In practical working conditions, the standard deviation of the $u$ component in the Kaimal wind spectrum
model undergoes certain variations due to spatial coherence influences.
(2) Spatial Coherence Model
To ensure consistency between calculation and actual conditions in establishing a three-dimensional
pulsating fluid field, the wind speed distribution across the entire swept area of the wind turbine cannot be
calculated using a single-point wind load spectrum alone; rather, consideration must be given to the mutual
relationship between various streamlines in space, which can be expressed through a spatial coherence model. The
IEC defines this relationship as the flow direction component fluid field coherence function (IEC 61400-3, 2001).
$$Coh_{i,j} = \exp\left(-a\sqrt{\left(\frac{f_r}{\overline{u}_{hub}}\right)^2 + \left(0.12\frac{r}{L_C}\right)^2}\right)$$
Where, $r$ is the distance between any point $i$ and $j$ on the grid; $f$ is the frequency; $L_C$ is the coherent scale
parameter; $\overline{u}_{hub}$ is the average wind speed at the wheel hub. The wind speed distribution in the fluid domain was
calculated, as shown in Figure 3.



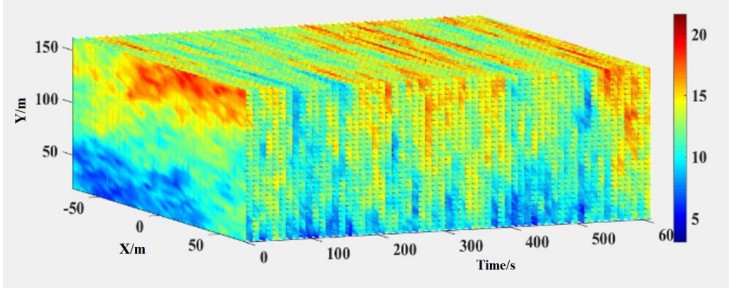


Fig. 3 Wind speed distribution in the computational fluid domain

**1.2 Calculation theory of EDEM ice load**

In contrast to other marine engineering structures, offshore wind turbines are characterized by their tall and
flexible design. When floating ice interacts with the tower in the presence of ocean currents and when the
frequency of ice loads matches the natural frequency of the overall wind turbine structure, it can lead to severe
structural response. This study employs the discrete element method along with Matlock single-tooth model and
asynchronous failure model to describe the interaction process between ice loads and wind turbine structures. The
Matlock single-tooth numerical calculation model assumes that each ice tooth undergoes linear elastic
deformation upon contact with the structure until reaching maximum deformation, after which point either no
longer contacting or breaking results in zero ice force. Thus, the expression for ice force is as follows:
$$F = \begin{cases} K_{ice}\Delta_i, & 0 < \Delta_i < \Delta_{max} \\ 0, & \Delta_i \leq 0, \Delta_i = \Delta_{max} \end{cases} \quad (3)$$

Where, $K_{ice}$ is ice tooth stiffness, N; $\Delta_i$ is the deformation of the $i$th ice tooth, m; $\Delta_{max}$ is the maximum
deformation of ice teeth, m. The floating ice calculation domain is shown in Figure 4.

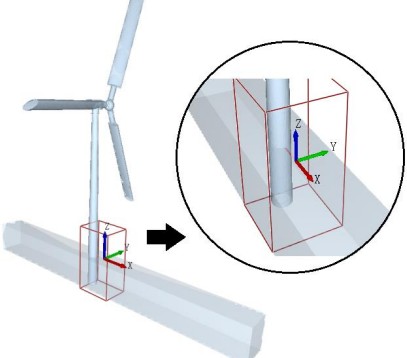


Fig. 4 Calculation domain of floating ice impact of wind turbine tower



In the asynchronous failure model, the ice tooth stiffness is determined based on the uniaxial compressive
strength of ice. Once the quasi-static ice force acting on the failure zone *i* reaches its limit, failure of the
corresponding ice element occurs. The total ice load at time *t* can be calculated as the summation of local ice loads
and can be expressed as follows:
$$F = \begin{cases} K_{ice}\left[ y + V_{ice}t - L(n-1)\right], & 0 < \Delta \le \Delta_L \\ 0, & \Delta \le 0 \end{cases} \qquad (4)$$

To enhance the precision of load calculation on wind turbine tower post floating ice impact, it is crucial to
emphasize the bonding bond (i.e., connection bond) among particles while establishing an ice model. The bonding
strength in the bonding bond is determined by the physical and chemical compositions of ice, ultimately
determining its structural type as depicted in Figure 5.

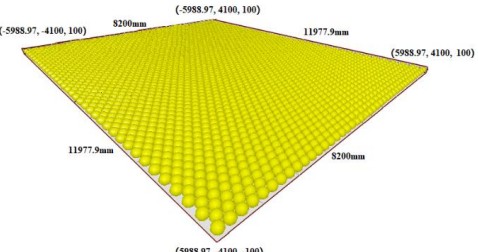


Fig. 5 Ice floe model

**2 Working condition simulation**
Wind turbines running in cold regions are subjected to a complex working environment, where wind load,
wave load, and ice load act together on the tower of wind turbines. The schematic diagram illustrating the
coupling of these loads is presented in Figure 6. In the calculation of wind load, a turbulent fluid domain is
established based on the Kaimal wind speed spectrum, and the momentum-leaf element theory is employed to
determine the wind load acting on offshore wind turbines.

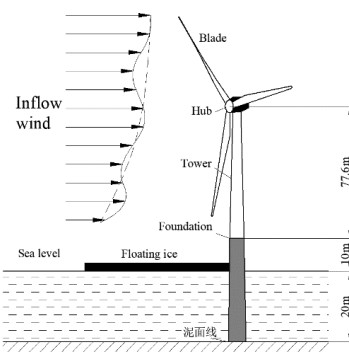




Fig. 6 Schematic diagram of loading of offshore wind turbines

**2.1 Turbulent fluid domain**

In the natural environment, air flow exhibits spatial non-uniformity and temporal unsteadiness. To capture

the statistical characteristics of wind speed time series in three directions, this study employs TurbSim, an open-
source turbulent wind stochastic simulator developed by NREL. The wind speed power spectrum in the frequency
domain is transformed into the time domain using inverse Fourier transform, enabling generation of three-
dimensional turbulent wind data for multiple points within the flow field domain based on spatial coherence
function. The structural dynamic response analysis and calculation of wind turbines are facilitated by calling the
wind data file during computation. Taking the NREL 5MW wind turbine as a case study, its key structural
parameters are presented in Table 1.

Tab. 1 Main parameters of NREL 5MW wind turbine

| Parameter | Value | Parameter name | Value |
|---|---|---|---|
| Rated power | 5MW | Cut-in wind speed | 25 m/s |
| Tower height | 87.6m | Cut-out wind speed | 3 m/s |
| Hub height | 90m | Rated wind speed | 11.4m/s |
| Hub diameter | 3m | Rotor quality | $1.11\times10^5$kg |
| Rotor diameter | 126m | Engine room mass | $2.4\times10^5$kg |
| Control system | Synchronous pitch | Tower mass | $3.48\times10^5$kg |

In the calculation process of OpenFAST, the fluid calculation domain's coverage area encompasses the entire

range of the wind turbine and tower. Therefore, the fluid calculation domain is divided into regions based on the
hub point. Considering the structural size data of the wind turbine and ensuring accurate calculations for
maximum tower deformation, a calculation interval within a 145m×145m range at hub center height is
determined. This interval is further divided into 15 grid regions for individual calculations. The flow field
calculation area's grid division can be seen in Figure 7, while maintaining the wind turbine's tower top coordinate
system as the basis for calculating coordinates.

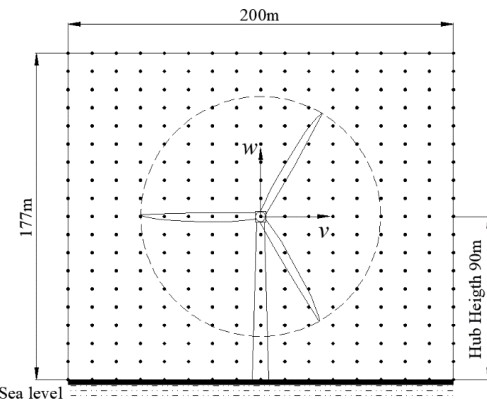


Fig. 7 Regional division of flow field

With the hub center as the reference point, based on relevant offshore wind resource data (LI Guanghua et al.,

2018), the time-domain average wind speed at the reference calculation point was determined to be 12m/s, with a
calculation time of 600s and a time step of 0.05s. By applying an inverse FFT transform to the NWTCUP wind
spectrum model (LI Chuangdi et al., 2019) and considering spatial coherence, we obtained the wind speed variation
characteristics at each grid node, which are depicted in Figure 8 showcasing the wind speed distribution within the
turbulent flow field calculation domain. As illustrated in Figure 8, it is evident that the wind speed within this flow
field undergoes iterative changes over time, exhibiting noticeable variations in vertical direction due to wind
shear.

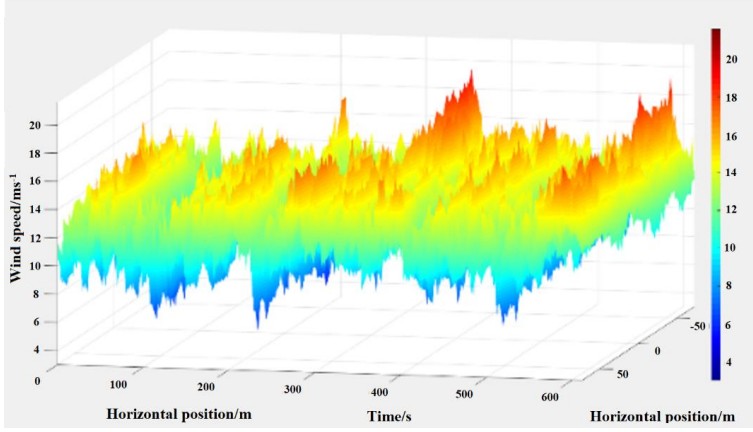


Fig. 8 Three-dimensional wind velocity distribution in the turbulent flow field calculation domain

**2.2 Selection of working conditions**

The effect of ice load on the tower foundation of wind turbines is primarily determined by various factors,

including ice speed, ice thickness, ice drift direction, and the movement of floating ice with ocean currents.




Therefore, the calculation of ice load involves considering wave load in a coupled superposition analysis. In line
with the research objectives of this study, we initially assume that wind load and ice load are independent from
each other. Additionally, we assume that the wind flow direction aligns with the drift direction of the ice and that
the running speed of ocean currents carrying ice is a key influencing factor for failure modes related to loads. The
failure modes associated with ice load can be categorized into three forms: intermittent extrusion fracture,
frequency self-locking fracture, and continuous extrusion fracture failure. Typically, when the ocean velocity in an
area covered by floating ice is less than or equal to $0.02\text{m}\cdot\text{s}^{-1}$ , intermittent extrusion fractures occur. When the
velocity falls within ($0.02\text{m}\cdot\text{s}^{-1}$, $0.04\text{m}\cdot\text{s}^{-1}$), frequency self-locking extrusion fractures are observed in floating ice
conditions; whereas velocities exceeding $0.04\text{m}\cdot\text{s}^{-1}$ result in continuous extrusion fractures in floating ice
scenarios. The calculation conditions for wind turbines under coupling effects between wind and ice loads are
presented in Table 2.

Tab. 2 Calculation conditions of offshore wind turbines

| No. | Ice load breaking type | Ice speed /m·s⁻¹ | Ice thickness /mm | Wind speed at hub height / m·s⁻¹ |
|---|---|---|---|---|
| 1 | / | / | / | 12.0 |
| 2 | Intermittent crushing | 0.01 | 12.5 | / |
| 3 | Frequency self-locking crushing | 0.021 | 12.5 | / |
| 4 | Continuous crushing | 0.05 | 12.5 | / |
| 5 | Intermittent crushing | 0.01 | 12.5 | 12.0 |
| 6 | Frequency self-locking crushing | 0.021 | 12.5 | 12.0 |
| 7 | Continuous crushing | 0.05 | 12.5 | 12.0 |

According to the conditions in the northern sea area of China (HUANG Lin et al., 2013), an ice thickness of
$h$=0.0125m was specified in the ice load boundary file. For different operational scenarios, values of $v_{ice}$=0.01ms⁻
¹, 0.021ms⁻¹, and 0.05ms⁻¹ were assigned for ice and ocean current velocities respectively. The structure diameter
$D$ was set at 4m, indentation coefficient $I_{km}$ at 2.7, ice brittleness strength σ=5MPa, ice teeth spacing $P$=1m, and
maximum elastic deflection $\Delta_{max}$=1m. Figure 9 illustrates the time history curve depicting the variation in ice load
under these three distinct working conditions.



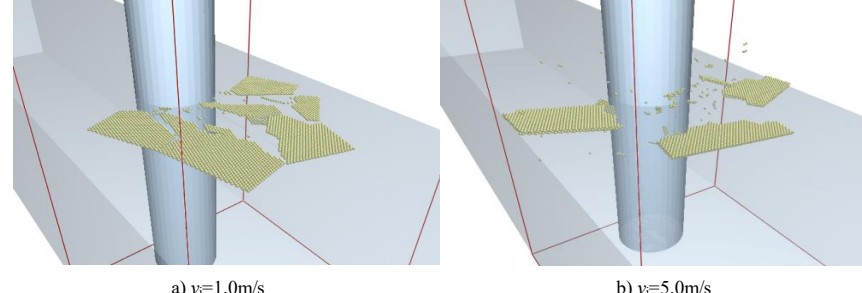

Fig. 9 The time history curve of ice loading

## 3 Engineering example calculation

### 3.1 Coupling effect of wave-ice load

This paper focuses on investigating the impact load and unit load of wind turbine towers subjected to
floating ice, while analyzing the influence of different running speeds and thicknesses of floating ice on these
towers. Figure 10 illustrates the occurrence of broken ice when the floating ice thickness is 12.5mm and the ice
speed is respectively 0.01m/s and 0.05m/s. Table 3 presents the impact load and unit load of wind turbine towers
for varying floating ice running speeds, namely, 0.01m/s, 0.021m/s, and 0.05m/s.

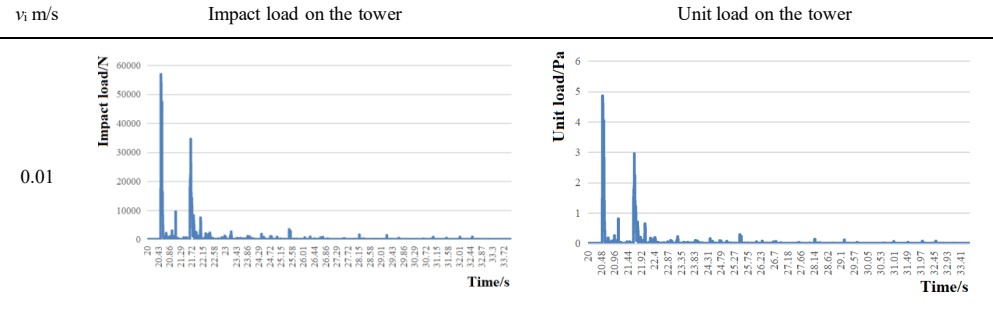

a) $v_i$=1.0m/s                                b) $v_i$=5.0m/s

Fig. 10 Fragmentation of floating ice with ice thickness of 12.5mm at different flow rates

Tab. 3 Ice thickness of 12.5mm, ice of speed 0.01m/s, 0.021m/s, 0.05m/s wind turbine tower load and unit load

| $v_i$ m/s | Impact load on the tower | Unit load on the tower |
|---|---|---|
| 0.01 | | |

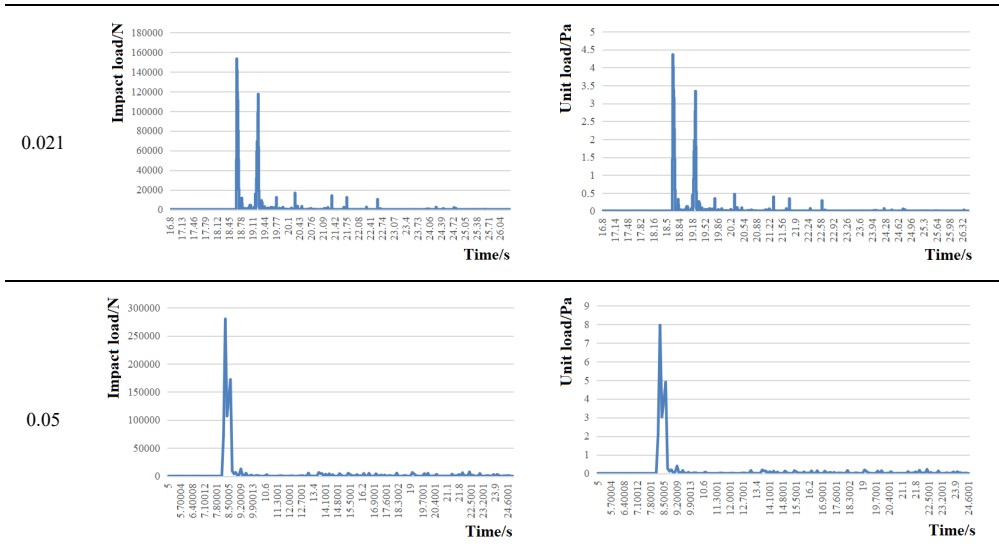

### 3.2 Coupling effect of wind-wave-ice load

The longitudinal displacement of the wind turbine tower top under the coupling effect of wind-wave-ice load, as shown in Fig.11a, is significantly greater than that under the coupling effect of wave-ice load alone. Moreover, the amplitude (displacement) of the tower top under the combined effect of all three loads is larger and more intense. Notably, when subjected to ice load alone, the vibration at the tower top fluctuates around its centerline with a rapid decay in amplitude from maximum to minimum after 80s. Conversely, under wind load alone or coupled with all three loads, the vibration equilibrium point at the tower top shifts approximately 0.3m along its longitudinal direction (i.e., x-direction in terms of tower top coordinate system). This indicates that when exposed to wind load, bending occurs along the flow direction and cannot be effectively recovered within this period. Additionally, when ice and wave loads are combined, there is an offset between their respective bending directions and resulting vibration equilibrium points at the tower. As depicted in Fig.11b regardless of variations in magnitude or direction for wind load conditions observed here, the vibration equilibrium point for longitudinal bending moment remains consistent with geometric shape center across different scenarios. Moreover, coupling effects among wind-wave-ice loads result in significantly greater vibrations compared to individual loading conditions. Furthermore, the shape and trend exhibited by longitudinal loading curves at foundation level remain consistent across these three working conditions, and it's worth noting that vibrational amplitude changes are considerably higher when all three loads are coupled compared to individual loading effects.

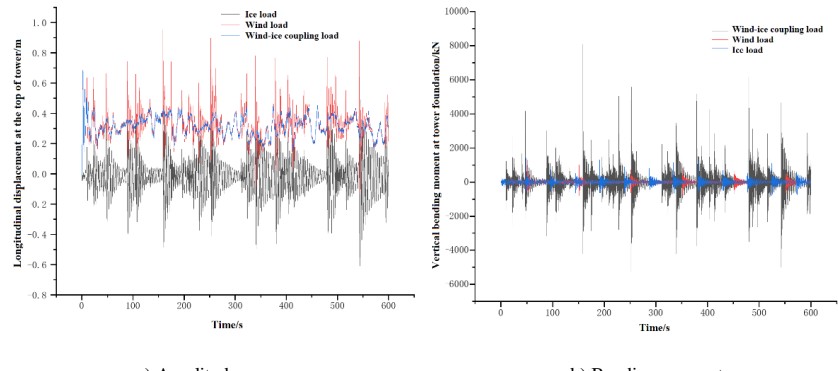


|         a) Amplitude        |        b) Bending moment        |

Fig. 11 Longitudinal physical quantities of offshore wind turbines under load

The longitudinal load and lateral load at the tower top of the wind turbine under the three-load coupling

effect are shown in Fig. 12a, while Fig. 12b illustrates the lateral bending moment and longitudinal bending
moment at the tower foundation under the same effect. Since the direction of bending moment is perpendicular to
that of load, i.e., transverse vector direction of tower load aligns with longitudinal vector direction of bending
moment, it can be observed from Figure that both load and bending moment exhibit vibrations around the
geometric center of the tower as their equilibrium point in its longitudinal direction. However, along the lateral
direction, both load and bending moment experience varying degrees of displacement from their respective
equilibrium points due to loads applied. Moreover, it is noteworthy that while there exists higher amplitude
variation in loads longitudinally compared to laterally with more random patterns; for bending moments, higher
amplitude variations occur laterally than longitudinally with greater randomness as well. Nevertheless, overall
randomness degree is found to be higher for loads than for bending moments.

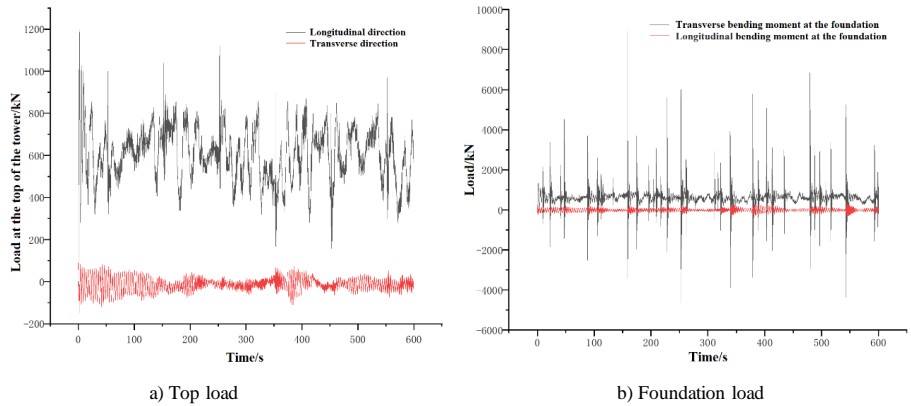


a) Top load                   b) Foundation load

Fig. 12 Load changes of offshore wind turbines

The lateral vibration amplitude of the support platform of wind turbines is generally greater than the

longitudinal vibration amplitude under various working conditions, as depicted in Figure 13. Moreover, the





longitudinal vibration amplitude remains small and exhibits overall stability, while the lateral vibration
displacement shows significant variability. Considering the impact of individual wind loads or ice loads, it is
observed that ice load has a higher effect compared to wind load. Furthermore, when considering coupling loads,
their effect surpasses that of any single load in terms of both tower amplitude after ice load and tower amplitude
attenuation frequency.

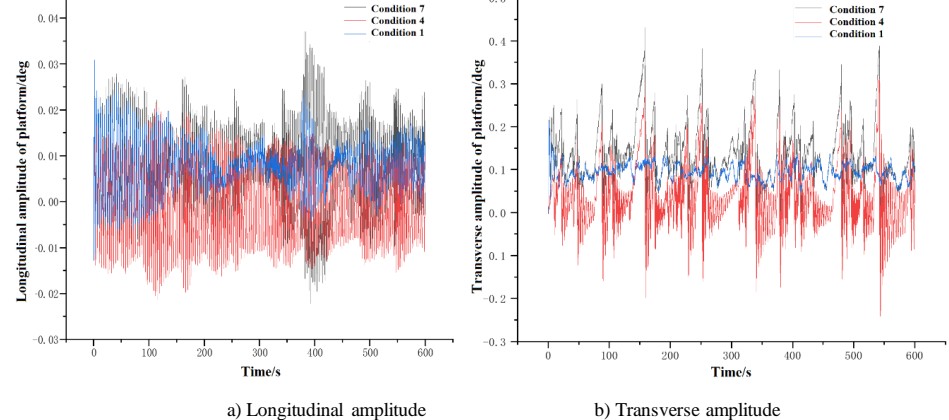

251      a) Longitudinal amplitude    b) Transverse amplitude

252        Fig.13 Amplitude of offshore wind turbine platform under load

253   The findings in Figure 14 demonstrate that, regardless of the running conditions of offshore wind turbines,

the vibration pattern and trend of the tower support platform align with those observed for tower vibration
displacement. Moreover, both in terms of amplitude and frequency, lateral load vibrations surpass longitudinal
vibrations.

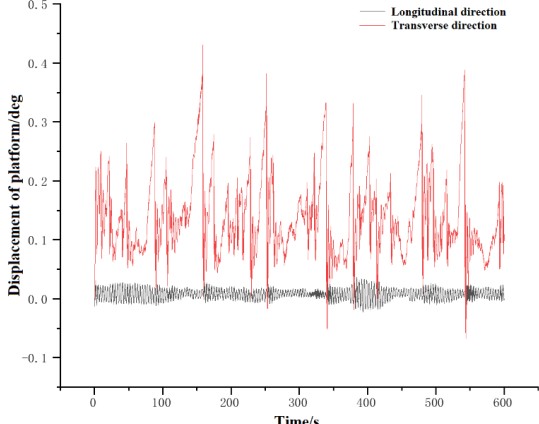

258     Fig. 14 Comprehensive vibration displacement changes of the platform under load

**3.3 Fatigue damage estimation**





The fatigue damage of wind turbines under the single action of wind load, ice load, and the coupling effect of
wind-wave-ice three loads is calculated in the time domain to analyze their long-term effects on load
accumulation. This study compares and analyzes the fatigue damage of offshore wind turbines under these three
working conditions using both the quadratic superposition method and DNV method for damage combination
calculation. As shown in Table 4, regardless of whether the quadratic superposition method or DNV method is
used for analysis and calculation, it is evident that the fatigue damage caused by ice load alone is lower than that
caused by wind load alone, while the fatigue damage under coupling loads is lower than that under single loads.
During wind-wave-ice coupling, results from the quadratic superposition method yield smaller calculations
compared to those from DNV's superposition method which yields larger but more accurate results suitable for
engineering applications in assessing fatigue damage of offshore wind turbines.
Tab.4 Fatigue damage of offshore wind turbines under different working conditions and different algorithms

| Wind load | Ice load | Load coupling | Quadratic superposition method | DNV method |
|---|---|---|---|---|
| $7.43\times10^{-8}$ | $5.23\times10^{-8}$ | $4.11\times10^{-8}$ | $9.12\times10^{-8}$ | $5.02\times10^{-7}$ |

## 4 Conclusion

The present study investigates the dynamic response of offshore wind turbines subjected to wind load, wave
load, ice load, and their coupled effects. Additionally, an estimation of fatigue damage is conducted. The obtained
results demonstrate that:
(1) Under the separate action of wind load and ice load, the horizontal and longitudinal bending moments at
the tower foundation of wind turbines are essentially identical. When wind-wave-ice load is coupled, the
horizontal and longitudinal bending moments at the tower foundation of wind turbines exceed those under
individual loads. Additionally, due to wind load participation, the equilibrium point for tower vibration in wind
turbines shifts along the direction of the wind load vector, with a linear translation corresponding to the calculated
wind speed. Under isolated ice load conditions, both horizontal and longitudinal amplitudes at the top of wind
turbines are smaller compared to those under isolated wind load conditions. This can be attributed to a longer
duration of action for wind loads despite their lower instantaneous impact force when compared to ice loads; thus
resulting in more pronounced cumulative effects. When considering coupled wind-wave-ice loading scenarios,
although similar trends are observed as seen under individual actions from winds and ice alone, there exist
significant differences in amplitude and attenuation rate. The displacement offset experienced by support
platforms in response to combined loading is greater than that caused by either winds or ice alone; moreover,
lateral displacement significantly exceeds longitudinal displacement on these platforms. Furthermore, it should be
noted that compared with its influence on tower foundations from winds alone, ice loads have a more noticeable
effect on displacements.
(2) When analyzing the impact of ice load on standalone wind turbines, the discrete element method is
employed to investigate the characteristics of floating ice impacting offshore wind turbine towers. To enhance the



accuracy in calculating the tower's impact load caused by ice, the mutual influence resulting from changes in
floating ice composition is achieved through designing numerical values for bonding bonds between particles.
This approach helps narrow the gap between simulation calculations and actual working conditions. The analysis
and calculation results reveal that the velocity at which floating ice moves significantly affects wind turbine
towers. Therefore, in practical projects, a well-designed tower foundation platform structure can be implemented
to reduce the speed at which floating ice moves, thereby mitigating damage caused by both floating ice and other
objects to supporting platforms of wind turbines while enhancing their structural stability and reliability.

(3) The fatigue damage value of wind load and ice load on wind turbines under separate action is greater than

the effect of coupling load, as the direction and magnitude of wind load are uncertain. This uncertainty helps to
balance the fatigue damage caused by ice load and wave load on wind turbines to some extent. In the calculation
process, the DNV method yields a higher damage value through superposition calculation compared to the
calculation result considering load coupling effects. From an engineering design and evaluation perspective, this
indicates a higher safety factor for the DNV method. Therefore, it is recommended to utilize the DNV method in
practical engineering applications for evaluating the fatigue life of offshore wind turbines under coupling loads.

**Competing interests:** The contact author has declared that none of the authors has any competing interests.

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
