# Peer review of "Dynamic Response of Offshore Wind Turbine Structure under Multi-load Coupling Based on DEM and FEM Joint Analysis"

_Wind Energy Science, 2024_

## Author Comment (AC1)

2024.9.9

*Wind Energy Science*

Manuscript ID: wes-2024-66

Manuscript title: Dynamic Response of Offshore Wind Turbine Structure under Multi-load Coupling Based on DEM and FEM Joint Analysis

Authors: Xin Guan, Haoran Xu, Ying Yuan, Shuaijie Wang, Chenhao Zhao, and Hua Yu

Dear Editor,

Thank you for your letter dated [2024.9.5]. We are pleased to know that our manuscript has been rated as potentially acceptable for publication in *Wind Energy Science*, subject to adequate revision and response to the comments raised by the reviewers.

Based on the instructions provided in the decision letter and comments provided by the reviewers, we have revised the manuscript, as shown below.

**Comment 1:** The literature study merely lists the use of methodologies by others without providing their findings and building upon them. A thread is also lacking. Also, the bibliography should be more diverse.

**Response:** In order to respect the research results of scholars, this paper only mentions the basic situation of his research results. In this paper, based on the research results completed by domestic and foreign researchers, combined with the load borne by the actual turbine operation, the DEM-FEM joint simulation calculation is used to directly describe the response of the offshore wind turbine under the load of floating ice.

**Comment 2:** The novelty aspects with respect to literature should be highlighted in the introduction/conclusion.

**Response:** The analysis method in this paper is the discrete element method, which can clearly show the dynamic change of the unit and the floating ice under the action of floating ice load, and has practical engineering value compared with the FEM dynamic load analysis only. Although this method has some limitations in analysis, it can be used as an engineering method for load calculation of offshore wind turbines under the action of floating ice.

**Comment 3:** Why did you choose the NREL 5MW for your work rather than newer and bigger concepts, like the DTU 10MW or the IEA 15MW that are the current reference in the community?

**Response:** 5MW offshore wind turbines are widely used in China's offshore wind farms, and most of the single installed capacity of wind turbines for offshore wind farms to be built or under construction is also NREL 5MW. The accuracy of the analysis can be verified by comparing the measured data of offshore wind turbines, and this research topic comes from the actual needs of enterprises. The purpose is mainly to solve the problem of engineering analysis of floating ice action of domestic offshore wind turbines, so NREL 5MW wind turbines are taken as the research object.

**Comment 4:** References to the tools and concepts you used are missing: OpenFAST is not cited; EDEM is not cited; NREL 5MW is not cited;

**Response:** According to your suggestion, I have finished supplementing the relevant reference documents.

**Comment 5:** The "Load calculation" section describes methodologies which are embedded in the engineering tools used. I suggest clarifying that it is either OpenFAST (uncited) or EDEM (uncited) that does the calculations. Also, specify for which output you use each tool. And also, what are you getting out of each tool you use?

**Response:** Combined with comment 4, the calculation results of OpenFAST and EDEM have been described respectively.

**Comment 6:** This methodology section is really missing how you are coupling the outcome of each tool. Did you couple the tools? Or did you just couple the results? And How?

**Response:** In this paper, the coupling interaction is carried out by software. The floating ice numerical model is established by DEM, and FEM is used as the flow carrier. The calculation step size is 0.5s. Firstly, the floating ice movement form when the first step size is 0.5s is calculated by FEM, and then the load effect of floating ice movement on the unit and the change of ocean current movement form are calculated by DEM. In the second step of 0.5s, FEM is used to calculate the first step time again, the movement form of ocean current and the movement change of floating ice, and so on, and the coupling calculation of the load change in the tower, the movement of floating ice and ocean current in the whole calculation period is completed.

**Comment 7:** What are the operating parameters of the turbine in the considered load cases? Also, how are you simulating the structure of the OWT? Are you accounting for aeroelasticity? Tower flexibility? Blade flexibility? If so, which are the elastic parameters? Is the turbine controlled and does it have an influence? – the wind turbine operation is not described at all except for the wind speed.

**Response:** This paper focuses on the study of the response of the wind turbine support structure under the action of ocean currents (carrying floating ice). If the operating state characteristics of the wind turbine are coupled, its operating characteristics are complicated, which is not the research scope of this paper. In consideration of the actual investigation, when the wind turbine is subjected to floating ice load, the control system does not participate in the control, and the wind turbine is in an uncontrolled state. The flexibility of tower and blade is estimated by setting the elastic modulus and Poisson's ratio of tower. In this paper, the dynamic response of wind turbine under the action of floating ice is studied. The operating state of its body has no obvious influence on the dynamic load response analysis, so the calculation can be ignored.

**Comment 8:** The title does not really convey the topic of the paper. The section titles should be revised as well.

**Response:** In response to your comments, the title has been partially modified.

**Comment 9:** Acronyms are not explained. FEM, DEM, DNV, …

**Response:** FEM, DEM, DNV, etc. have been explained

**Comment 10:** Figure labels are often too small

**Response:** The font size of the label text in the Figure has been changed

**Comment 11:** The introduction section should be 1 and not 0

**Response:** The format has been modified according to the review expert opinion.

**Comment 12:** Author contributions are missing

**Response:** Contributions to the work of each author have been supplemented.

We would like to take this opportunity to express our sincere thanks to the reviewers who identified areas of the manuscript that needed corrections or modification. We would like also to thank you for allowing us to resubmit a revised copy of the manuscript.

We hope that the revised manuscript is accepted for publication in *Wind Energy Science*.

Sincerely Yours,

Xin Guan

Shenyang Institute of Engineering

Shenyang 110136, China

Phone No: 13840506103

Email Address: xin_guan@sina.com; xhrlzs@163.com

---

## Author Comment (AC2)

2024.9.13

*Wind Energy Science*

Manuscript ID: wes-2024-66

Manuscript title: Dynamic Response of Offshore Wind Turbine Structure under Multi-load Coupling Based on DEM and FEM Joint Analysis

Authors: Xin Guan, Haoran Xu, Ying Yuan, Shuaijie Wang, Chenhao Zhao, and Hua Yu

Dear Editor,

Thank you for your letter dated [2024.9.5]. We are pleased to know that our manuscript has been rated as potentially acceptable for publication in *Wind Energy Science*, subject to adequate revision and response to the comments raised by the reviewers.

Based on the instructions provided in the decision letter and comments provided by the reviewers, we have revised the manuscript, as shown below.

**Comment 1:**

It is not obvious to the reviewer how this work relates to the existing literature as well as where the incremental innovation comes out of this work. It is also not clear if the authors are developing mathematical models to analyse the coupled loading (wind, wave, ice) or if they are using the already established models to analyse their impact on accumulated fatigue damage.

**Response:**

Because there are few literatures on the dynamic response of offshore wind turbines under the coupling of wind load, wave load and floating ice load, and the analysis of single load is the main one. In discussing the existing literature achievements, we only explain the main contents of the scholars' research, and use some of the research contents as the research basis of this research. This paper is mainly an engineering method to explore the dynamic response of offshore wind turbines under multiple loads (wind, wave, ice), and study the established load model for the cumulative fatigue damage analysis of offshore wind turbines.

**Comment 2:**

Is Figure 3 produced by the authors? If not, please provide a ref.

**Response:**

In this paper, the published random wind speed data and the calculation results of formula 3 were used as data sources to form wind load spectrum through OpenFAST simulation (Figure 3).

**Comment 3:**

There is no detailed explanation of the ice load calculation. How can one evaluate the validity considering that there is a reference to the explicit nonlinear structural response analysis?

**Response:**

By referring to the relevant literature of Marine science, the floating ice types that appear more frequently in the freezing period in the cold area are obtained and taken as the object of analysis. This study only discusses the analysis method of the dynamic response of offshore wind turbines under multi-load coupling. If it is widely recognized by the wind power engineering, it can be applied to the actual offshore wind power projects. The specific floating ice analysis model can be adjusted by DEM according to the actual ice type data to make it conform to the specific actual characteristics of offshore wind power projects.

**Comment 4:**

The reviewer is having a difficult time understanding why the authors are illustrating Figure 8 if there is no contribution from the authors to get this 3D figure.

**Response:**

FIG. 8 is not an illustration of 3D diagram 3, but the load diagram of offshore wind turbines after multi-load (wind, wave, ice) coupling is used as the boundary condition for subsequent multi-load coupling analysis.

**Comment 5:**

Any references from Table 2?

**Response:**

Tab. 2 shows the analysis results of multi-load coupled offshore wind turbines calculated by OpenFAST, instead of referring to other literature.

We would like to take this opportunity to express our sincere thanks to the reviewers who identified areas of the manuscript that needed corrections or modification. We would like also to thank you for allowing us to resubmit a revised copy of the manuscript.

We hope that the revised manuscript is accepted for publication in *Wind Energy Science*.

Sincerely Yours,

Xin Guan

Shenyang Institute of Engineering

Shenyang 110136, China

Phone No: 13840506103

Email Address: xin_guan@sina.com; xhrlzs@163.com